# Optimization and Scale-Up of Fermentation Processes Driven by Models

**DOI:** 10.3390/bioengineering9090473

**Published:** 2022-09-14

**Authors:** Yuan-Hang Du, Min-Yu Wang, Lin-Hui Yang, Ling-Ling Tong, Dong-Sheng Guo, Xiao-Jun Ji

**Affiliations:** 1School of Food Science and Pharmaceutical Engineering, Nanjing Normal University, Nanjing 210023, China; 2State Key Laboratory of Materials-Oriented Chemical Engineering, College of Biotechnology and Pharmaceutical Engineering, Nanjing Tech University, Nanjing 211816, China

**Keywords:** mechanistic modeling, data-driven, hybrid modeling, scale-up, computational fluid dynamics

## Abstract

In the era of sustainable development, the use of cell factories to produce various compounds by fermentation has attracted extensive attention; however, industrial fermentation requires not only efficient production strains, but also suitable extracellular conditions and medium components, as well as scaling-up. In this regard, the use of biological models has received much attention, and this review will provide guidance for the rapid selection of biological models. This paper first introduces two mechanistic modeling methods, kinetic modeling and constraint-based modeling (CBM), and generalizes their applications in practice. Next, we review data-driven modeling based on machine learning (ML), and highlight the application scope of different learning algorithms. The combined use of ML and CBM for constructing hybrid models is further discussed. At the end, we also discuss the recent strategies for predicting bioreactor scale-up and culture behavior through a combination of biological models and computational fluid dynamics (CFD) models.

## 1. Introduction

With the increasing consumption of fossil fuels and related environmental issues, there is an urgent need to find biological substitutes for traditional petrochemical products through green biological manufacturing. Over the decades, microorganisms have been used as ‘mini-factories’ in biomanufacturing to aid with the diversity of metabolic pathways and their accompanying ability to transform a wide range of renewable raw materials into value-added compounds by means of fermentation [1,2,3]; however, the use of wild-type microorganisms in industrial production is usually affected by many factors, such as substrate and product toxicity [4,5]. Biologists are provided with many modern biotechnologies, such as genetic engineering and synthetic biology, to engineer more powerful mini-factories [6,7,8,9,10]. Successful examples include *Escherichia coli*, which produce insulin and carotene [11,12], *Saccharomyces cerevisiae*, which produce geraniol [13], *Yarrowia lipolytica*, which produce N-acetylneuraminic acid [14], *Bacillus subtilis*, which produce hyaluronic acid [15], and so on; however, the construction of mini-factories is not the end goal of biomanufacturing—biological fermentation and industrial production are the ultimate goal, which is not a simple task. The engineered strains may not perform well in the actual fermentation process due to the lack of suitable fermentation strategies; therefore, optimizing fermentation parameters (such as the medium composition and extracellular conditions) is a key factor in terms of running the mini-factories efficiently, which is crucial for the process of fermentation [16,17,18]. In practice, the rich insights generated by mathematical modeling can assist in the optimization of fermentation processes [19]. Mathematical models, as approximations of reality, can clearly represent fermentation processes whose intrinsic complexity exceeds intuitive understanding, thus providing indispensable insight into designing, controlling, and optimizing the process, as well as minimizing unnecessary experimentation [20,21].

However, it is not easy to model the fermentation process because each cell in the bioreactor can be viewed as a subsystem of metabolic and signaling networks [22,23]. For fermentation problems, three modeling approaches are generally used: mechanistic modeling, data-driven modeling, and hybrid modeling [20,24]. The mechanistic modeling approaches derive the models from prior knowledge using equations that are notable and acknowledged. Mechanistic models can extract valuable information from the raw data, and they provide insight into the underlying mechanisms [25]. Kinetics and constraint-based modeling (CBM) are the two primary mechanistic approaches for analyzing microbial growth and metabolism [26]. In contrast, the data-driven approaches obtain a model by analyzing and fitting existing data. Data-driven models are also known as black-box models because they cannot provide information about the basic mechanisms without considering their internal structures and phenomena [27]. Machine learning (ML) is a commonly used data-driven approach. With the advancement of omics technology and various analytical techniques, the datasets available for fermentation process modeling are rapidly growing, and academics favor using ML to interpret large-scale datasets for deeper analysis and optimization [28]. As a result of such circumstances, hybrid modeling has come into being, which refers to the integration of mechanistic modeling and data-driven approaches. Recently, comprehensive reviews of hybrid models have been published, indicating that they are a promising prospect for this field [29,30].

In addition to the complexity of microbial metabolic behavior, fermentation systems also have complex hierarchical structures. These systems consist of microorganisms and a fermentation environment, and they are influenced by upstream and downstream operating conditions. Moreover, during industrialization, the expansion of the bioreactor volume and changes in shape lead to changes in the fermentation environment, which subsequently leads to the failure of fermentation strategies that are developed during the laboratory stage [31]; therefore, as biological fermentation shifts from laboratory to industrial production, we also need to introduce fermentation environmental changes into biological models to elucidate the effects of mixing and hydrodynamics [32,33]. This goal can be achieved by coupling biological models with computational fluid dynamics (CFD) models.

This paper gives an overview of different mathematical modeling methods and their applications in biological fermentation processes. We first introduce the basic forms of mechanistic models that can describe microbial metabolism using kinetic and CBM modeling methods and their applications in biological fermentation processes. Next, we discuss different approaches to building data-driven models using ML. The synergistic effect of CBM and ML is further discussed. In the end, we highlight the coupling of biological models with CFD models, which prompts the formation of model-based integrated tools to successfully predict bioreactor scale-up and culture behavior during model-assisted bioreactor operation design.

## 2. Methods and Applications of Mechanistic Modeling

Mechanistic modeling describes some, though not all, mechanisms of complex systems, enabling model parameters to be measured or inferred. Kinetic models are a kind of common mechanistic model, which usually reveal the dynamic changes between metabolites by means of kinetic laws expressed by ordinary differential equations (ODEs) [34]; however, for kinetic models, the mechanism of gene regulation underlying biological phenomena is usually only explained by kinetic parameters in model equations (e.g., enzyme constants and metabolite concentrations), rather than a mechanistic description of gene action [35], and therefore, kinetic modeling is generally applied to fermentation problems in which the metabolic behavior is well understood. CBM is another mechanistic modeling approach. CBM studies the behavior of genome-wide systems to elucidate the relationship between genotypes, phenotypes, and environmental conditions [27]. Next, we will discuss these two mechanistic modeling methods and their application to fermentation problems in detail.

### 2.1. Application of Kinetic Modeling to Fermentation Processes

The derivation of kinetic models depends mainly upon the level of detail in which microbial growth and metabolic behaviors are described. From a macro perspective, the fermentation process in a bioreactor is a chemical process affected by physical and chemical environmental factors (such as temperature, pH, aeration, and substrate concentration). The transformation of substrate S to product P is catalyzed by bacterial pellets X, which may be seen as a box containing equally disseminated catalysts and chemicals. In this case, we do not need to pay attention to the internal structure of bacterial pellets; we only need to pay attention to the changes in substances and external environmental factors before and after catalysis. Thus, the kinetic models provide a macro perspective on the bioreactor dynamics without specifying compositional and structural details under reasonable assumptions. The kinetic models constructed in this case can also be called unstructured kinetic models. When the substrate concentration (S) is the only limiting element, the Monod model is established. Different unstructured kinetic models of fermentation processes are established with different descriptions of constraints and common kinetic equations, as shown in Table 1. In practice, biologists construct kinetic models by adjusting and combining different equations according to the actual situation. For instance, Zhang et al., established a new kinetic model based on Monod through parameter optimization, which not only described the relationship between products and by-products in the fermentation process of 1, 3-propylene glycol, but it also guided the optimization of the fermentation conditions and improved the yield [36]. Moreover, Garnier et al. developed an analytical solution that combined the Monod model with the Luedeking–Piret equation which can be used to simulate the batch fermentation process [37]. Although these models address the least complex portrayal of cellular behavior, they can prove particularly valuable when quantifying the phenomenon of growth inhibition during early process development. On the other hand, unstructured kinetic models generally contain few parameters, which, from a computational perspective, is usually advantageous. At present, a significant portion of recent studies concern bioethanol production [38,39,40], biosurfactant production [41,42], bacterial and fungal biomass production [43,44], as well as microalgal growth [45,46,47].

As opposed to unstructured kinetic models, structured kinetic models consider the physiological state of the cell and treat bacterial pellets as multicomponent entities with internal structures. Structure, in structured kinetic models, can be introduced by lumping the metabolites into distinct intracellular pools [25]. Wang et al. constructed a structured kinetic model that can be used to dynamically reflect changes in microalgal biomass by analyzing how carbon is allocated to different intracellular compartments under nitrogen-poor and nitrogen-rich extracellular conditions [56]. In a similar manner, Haringa et al. constructed a structured kinetic model describing the growth of *Penicillium*
*chrysogenum*, as well as product formation, by grouping the most important intracellular metabolites into five pools and four intracellular enzyme pools [57]. Their research also proved that the structured models have a higher accuracy and wider application scope than the unstructured models. We believe that our improved understanding of the behavior of microbial cells, along with advances in various omics techniques, will facilitate the further development of structured kinetic modeling.

### 2.2. Application of Constraint-Based Modeling to Fermentation Processes

CBM is another important mechanistic modeling method that considers the underlying metabolic mechanisms of microorganisms. CBM can be used to analyze the dynamic characteristics of specific target metabolites, and then implement fermentation strategy optimization from the micro-level, which can enhance the diversity and pertinence of optimization target design [58]. To build a CBM model, an in-depth understanding of metabolism is first required. Metabolism represents all the biochemical reactions in cells. These reactions form a coordinated set of metabolic pathways that convert substrates into the metabolites required for the physiological activities of the cells at a rate known as metabolic flux. In addition to being a basic determinant of a cellular metabolic state and physiology, metabolic flux is also the key parameter in quantifying metabolic networks [59]. CBM plays an important role in the metabolic modeling of quantitative metabolic flux [60,61,62]. Metabolic flux analysis (MFA) and Flux balance analysis (FBA) are the two primary CBM strategies for quantifying metabolic flux (Figure 1), which can be implemented by COnstraint-Based Reconstruction and Analysis (COBRApy) in Python [26,63].

#### 2.2.1. Flux Balance Analysis

Flux balance analysis (FBA) is a mathematical metabolic flux technique that analyzes the flux of metabolites through a detailed metabolic network (e.g., genomic scale) [64]. One of the earliest applications of FBA was used to calculate the given network model and substrate to obtain their maximum theoretical yields of metabolic products (Table 2) [64,65]. Papoutsakis et al. first demonstrated this in 1984 by describing the interrelationship between various products and biomass in butyrate fermentation using FBA, accurately predicting the theoretical yield of the several fermentation products of butyrate bacteria [66]. The same method can be utilized to work out the maximum biomass yield. For instance, Varma and Palsson used FBA to quantitatively anticipate the maximum cell growth rate cell density of wild-type *E. coli* W3110 [67]. In their study, they also used FBA to predict the temporal distribution of glucose as well as the concentration of by-products. Another important application of FBA focuses on the prediction of biomass production under different substrate conditions, which aims to improve medium formulation and/or medium feeding strategies [63]. Swayambhu et al. identified amino acids and carbon sources with significant effects on yield, and they further improved the yield of target compounds in recombinant *E. coli* by using medium optimization through FBA [68]. Kaushal et al. used FBA to study the effect of medium components on the metabolic processes in the simultaneous production of ethanol and butanol by *Clostridium sporogenes* NCIM 2918, which provided ideas for the optimization of butyric acid and ethanol fermentation [69]. Huang et al. also used FBA to compare the catabolism of Chinese hamster ovary (CHO) cells under different feed conditions in order to optimize medium formulations and increase antibody production [70]; however, FBA also has disadvantages, such as poor performance in predicting metabolic fluxes and growth phenotypes of engineered strains. In their study, Long and Antoniewic et al. demonstrated that FBA is a poor predictor of growth rate and metabolic flux in knockout strains [71,72]. Several complementary algorithmic approaches have been created to aid the likelihood of foreseeing the flux of genetically engineered strains, including regulatory on/off minimization (ROOM) [73], minimization of metabolic adjustment (MOMA) [74], and the relative change (RELATCH) algorithms [75]; MFA can compensate for this shortcoming.

#### 2.2.2. Metabolic Flux Analysis

Metabolic flux analysis (MFA) is an imaging flux omics technique that determines metabolic flux distributions by analyzing metabolic product production and consumption rates in biological systems [76]. Different from FBA, MFA focuses on studying the metabolic flux in cells under different environmental conditions without paying attention to the theoretical optimal solution [76]. For instance, Calik et al. used MFA to study the effect of pH on the metabolic flux of *Bacillus licheniformis*, and they proposed a pH manipulation strategy to improve the yield of β-lactamase [77]. Sarma et al. used MFA to conduct in-depth research into the enhancing effect of ultrasound-induced biohydrogen production, and they provided ideas for strategy optimization [78]. Xiong et al. also used MFA to quantitatively analyze metabolic flux during the production of L-lactate from glucose in *Lactobacillus*, and they proposed a temperature control strategy that could maximize the yield of L-lactate [79]. Another common application of MFA is to analyze the production of key cellular co-factors under different growth conditions. Li et al. used MFA to analyze the uneconomical utilization of ATP in soybean oil feeding *Acremonium chrysogenum*, and they proposed strategies to improve cephalosporin C (CPC) based on controlling NADPH and ATP production and consumption [80]. In addition, MFA can predict changes in metabolic flux in metabolically engineered strains, which provides help when selecting appropriate metabolic engineering strategies [81]. At present, MFA has been successfully used in the performance optimization of many industrial strains. [82,83,84,85,86]. Of course, MFA also has its disadvantages. One limitation is that the metabolic network used for MFA analysis often needs to be reasonably simplified; thus, some scholars have proposed optimization algorithms based on MFA, such as flux variability analysis (FVA), which not only improve the accuracy of MFA, but also broaden the application range of MFA [87,88].

#### 2.2.3. Dynamic Flux Balance Analysis

It is worth noting that MFA and FBA consider the external environment to be in a steady state, and they ignore the kinetics of enzymatic reactions during the modeling process. Dynamic flux balance analysis (DFBA) was produced by considering the relationship between the macro-state parameters of the fermentation process and the number of cell physiological metabolic parameters [89]. Recent studies include predicting shikimic acid production in *E. coli* [90], ethanol production in *Saccharomyces cerevisiae* [91], and the overproduction of secretory proteins in *Streptomyces lividans* [92]. DFBA can be constructed in conjunction with macro kinetics; for example, Henson et al. combined the Monod equation for substrate absorption with FBA to account for missing regulatory mechanisms [93]. In addition to being combined with kinetics equations, DFBA can also be implemented using ML in the form of hybrid modeling [35].

**Table 2 bioengineering-09-00473-t002:** Overview of CBM applications for the analysis and optimization of the fermentation parameters demonstrated in this review.

Parameter	Approach	Case	Refs.
Theoretical maximum	FBA	The relationship between various products and biomass in the process of butyric acid fermentation was described, and the theoretical yield of several fermentation products of butyric acid bacteria was predicted accurately.	[66]
Theoretical maximum	FBA	Quantitative prediction of maximum cell growth rate and cell density of wild-type *E. coli* W3110 were clarified.	[67]
Culture medium	FBA	Amino acids and carbon sources that have a significant influence on the yield were identified, and the yield of siderophore compounds in recombinant *E. coli* was improved by medium optimization.	[68]
Culture medium	FBA	The effects of glucose, glycerol, and the mixture of glucose and glycerol on the distribution of carbon flux in the simultaneous production of ethanol and butanol by *Clostridium sporogenes* NCIM 2918 were studied.	[69]
Culture medium	FBA	The effects of amino acid composition in a culture medium on the catabolism of Chinese hamster ovary (CHO) cells were analyzed to optimize culture medium formulation and increase antibody production.	[70]
pH	MFA	By analyzing the effect of pH on the intracellular metabolic network of β-lactamase producing *Bacillus licheniformis*, a pH manipulation strategy was proposed to improve the yield of β -lactamase.	[77]
Ultrasound	MFA	The effect of ultrasound promoting biological hydrogen production from glycerol fermentation was understood to a significant extent, and an optimal strategy of enhancing glycerol uptake and blocking the butyric acid pathway under the guidance of the MFA model was proposed.	[78]
Temperature	MFA	By quantifying the flux during l-lactic acid production from glucose, a temperature control strategy was proposed to maximize the productivity of L-lactic acid.	[79]

## 3. Methods and Applications of Data-Driven Modeling

To fully describe the nonlinear changes of the fermentation process, more parameters and state variables need to be used to build the mechanistic models [94,95]. Building such a model is not always a viable option because parameter determination requires complex experiments and large calculations [35]. Fortunately, advances in computer technology have facilitated the emergence of data-driven modeling, which can provide a suitable solution to this problem [96,97]. Figure 2 illustrates the modeling process for ML, a popular data-driven modeling technique. ML uses mathematical knowledge such as statistical theory to analyze datasets and find hidden relationships between existing data to justify phenomena [96]. Unlike mechanistic models, data-driven models created by ML are built on mathematical expressions, and model parameters have no physical, chemical, or biological significance. [28]. Data-driven models reflect mappings between data rather than causal relationships, so they tend to perform poorly when conditions change. Nonetheless, if relevant data is readily available, data-driven modeling can save a lot of time compared with mechanistic modeling [27]. Depending on the algorithm, ML can be categorized into two groups: supervised ML and unsupervised ML.

### 3.1. Supervised Machine Learning

The data points for a supervised learning algorithm each have an associated feature (input) value and a matching target (output) value. The feature value is the original measurement data (e.g., temperature, pH), and the target value is mainly composed of the reference data of process variables (e.g., productivity); therefore, each data point includes the numerical values or classes for each feature and desired value. The goal of a supervised learning algorithm is to mine the correlation between the feature and the target values in the dataset, and then predict the target value by using the new input feature value or find the optimal feature value by using the required target value [96].

The support vector machines algorithm (SVM) is a common supervised learning algorithm and can be trained using various packages, including Scikit-learn [98] in Python, Caret [99] in R, and MLJ [100] in Julia. The SVM can realize data classification and regression, and it can determine the best operating conditions by integrating with orthogonal experimental design [101]. Dong et al. analyzed the comprehensive effects of corn stalk weight, ultrasonic duration time, acoustic frequency, and alkali pretreatment time on biogas production, based on the SVM model [102]. Under the guidance of the model, they further obtained the optimal operating conditions and improved the gas production and efficiency of corn straw anaerobic fermentation. Zhang et al. also optimized the fermentation conditions of *Rhodotorula glutinis* using ethanol wastewater to produce biological lipids by constructing a SVM model, and they improved the biomass and lipid value yield [103]. In addition, the SVM is often used to classify and annotate the structure and function of proteins and genes due to its powerful regression and classification functions [104,105,106]. Artificial neural networks (ANNs) are another supervised learning algorithm that performs better than SVM when the dataset is large [107,108]. The software programs PyTorch [109] and Tensorflow [110] are frequently used to train neural networks. At present, ANNs have been successfully used in many studies on fermentation prediction and optimization. For example, Melcher et al. built a batch fermentation model of *E. coli* using ANNs, and they successfully predicted the concentration of biomass and recombinant protein [111]. In a recent study, Wang et al. also predicted the bio-hydrogen fermentation conditions and analyzed the effect of critical parameters during fermentation by combining ANNs and a response surface methodology [112]. ANNs also show great potential during fermentation process control. Recently, Tavasoli et al. used an ANN to dynamically regulate the fed-batch fermentation of recombinant *Pichia pastoris*, which significantly increased the yield of alpha 1-antitrypsin [113]. In addition, ANNs can also be used to assist processing massive omics data and parsing microbial metabolic mechanisms [114,115,116].

### 3.2. Unsupervised Machine Learning

Unsupervised learning algorithms, which aim to find hidden relationships and clusters, or to detect outliers, are suitable for datasets consisting of data points with only feature values [24]. Common unsupervised learning algorithms include dimensionality reduction algorithms and clustering algorithms [96].

The dimensionality reduction algorithms delete some unimportant data, and they discover the hidden relationship between data while preserving as much of the relationships between the data points as possible [96]. Specific methods include principal component analysis (PCA), principal component regression (PCR), and partial least squares regression (PLSR) [117]. At present, the algorithms have been widely used in the analysis of fermentation process. For instance, Gutiérrez-González et al. performed PCA to test whether there was any relationship between protein expression and favored environmental factors [118]. The results showed that temperature and reaction time had different effects on soluble and insoluble proteins. As another unsupervised learning algorithm, the clustering algorithm is widely used in biology. Arian et al. used the k-nearest neighbor algorithm, which is a clustering algorithm, to cluster protein kinases according to their activity [119]. In addition, unsupervised learning can also be used as a data preprocessing method during supervised learning (e.g., dimensionality reduction for a dataset used for supervised learning or the creation of category features).

## 4. Hybrid Models and Modeling Methods

Even for the most intensively studied model organisms, developing comprehensive mechanistic models is still an ambitious goal. It is an ideal solution to use ML’s powerful data processing and analysis capabilities to construct CBM models for elucidating metabolic mechanisms, so hybrid models that combine data-driven and mechanistic models are expected. Hybrid models have attracted much attention because they combine the accuracy of mechanistic models constructed using CBM with the efficiency of data-driven models constructed using ML [120,121]. There are three main methods for ML and CBM integration [122] (Figure 3):

(a)CBM models were constructed using ML by merging and analyzing omics data from different sources, whereas CBM was trained and reassembled by obtaining genomic data under specific conditions [123]. This method is suitable for situations wherein mechanistic models are not accurate enough. Vijayakumar et al. used ML to analyze RNA sequencing data extracted under 23 different growth conditions, which was then combined with flux data obtained by using FBA to elucidate the mechanisms underlying cyanobacterial responses to fluctuations in light intensity and salinity [124]. The growth rates of yeasts, such as *S. cerevisiae,* have also been predicted using this technique. For example, Culley et al. obtained reliable results for the growth rate prediction of *S. cerevisiae* by combining CBM-derived flux omics data with transcriptomics using ANNs [116].(b)Metabolic flux data obtained from CBM was trained by the ML method to gain more biological insights into the required system [125]. With this method, potential phenomena that cannot be mechanistically described can be analyzed. For example, Sridhara et al. used ML to analyze the metabolic flux data generated by CBM, and they realized that the retroversion of the culture medium components, which occurred during bacterial growth, could not be achieved using CBM alone [126].(c)ML can be used to analyze multi-omics data so as to provide data preprocessing services for CBM model construction. In 2016, Wu et al. used the ML method to analyze and integrate heterotrophic bacterial metabolic data from about 100 papers, finally constructing MFlux, a Web-based platform that can analyze metabolic fluxes [127].

As stated above, more successful mechanistic models describing metabolic dynamics can be created with a thorough understanding of microbial metabolism and the development of CBM and ML methods [128]; however, in order to accurately describe the fermentation process and predict the fermentation results, in addition to the metabolic models of the cells, the fermentation environment in the bioreactor also needs to be considered [25].

## 5. Coupling Biological Models with Computational Fluid Dynamics Models Enabling Rational Fermentation Scale-Up

The physiology of microorganisms and the flow field in the bioreactor are two important issues that are highly interrelated throughout the fermentation process, and they affect the final fermentation yield. The exploration into ideal genetic engineering modification strategies and fermentation conditions based on mechanistic and data-driven models enable efficient biological production on a laboratory scale; however, scaling-up fermentation processes from the laboratory to industrial production level requires more work. In industrial bioreactors, the spatiotemporal gradients of substrate, temperature, and pH can affect the metabolism of the resulting strains [39,129]. In addition, strong agitation in industrial bioreactors may also cause shear damage to cells and reduce their viability [130]. These factors lead to a loss of performance in scale-up operations, including a reduction in titer, yield, or productivity, as well as potential pitfalls for process monitoring and product quality control [131,132,133]; therefore, to narrow the gap from theory to practice, and to apply biological models in order to aid scaling-up production, we need to consider the impact of external environmental changes on microorganisms during modeling. CFD is a combination of mathematics, fluid mechanics, and computer technology that can be used to generate detailed flow field information in a bioreactor [134,135,136]; therefore, coupling the flow field information obtained by CFD models with the cell metabolism information provided by biological models can help to clarify how turbulence and environmental gradients occur, and ultimately, how they affect cell performance, which will provide guidance for bioreactor design and scaling-up production (Figure 4) [137]. For the purposes of our analysis, the biomaterial is referred to as the “biotic phase” and the culture environment is referred to as the “abiotic phase”. In the process of coupling biological models with CFD models, there are two main ways according to the different treatment methods of biotic phase: Euler–Lagrange methodology (ELM) and Euler–Euler methodology (EEM) (Table 3).

The ELM considers the biotic phase to be the period wherein discrete particles shuttle through the continuous abiotic phase, calculating each cell’s time-dependent positions in the main field in order to track their movement and state [137,138]. The ELM is suitable for coupling structured kinetic models in order to explain the differences between cell metabolisms, which are caused by environmental differences [139]. The ELM can be used to quantitatively analyze the effects of the concentration gradient on microorganism cultures [133]. For example, Siebler et al. investigated the effect of different concentration field residence times on transcriptional changes in *Clostridium ljungdahlii* cells under stress conditions during syngas fermentation [140]. Haringa et al. used CFD simulations of industrial bioreactors in conjunction with structured kinetic models to evaluate the impact of glucose concentration gradients on penicillin production [57]. Kuschel et al. also predicted differences between the metabolic statuses of cell populations due to glucose concentration gradients using the ELM [141]. For microalgae cultures, light intensity is an important factor. The growth of microalgae can be better predicted by tracking the light changes experienced by each cell, which can also be achieved using the ELM [142]. Gernigon et al. used the ELM to simulate the light distribution in two photobioreactors with different structures to analyze the influence of light on microalgae culture [143]. The results show that different light distributions can lead to different growth rates in microalgae, even when using the same light intensity; however, the ELM is computationally expensive and difficult to be applied on an industrial scale because it requires calculating change in each particle.

Another CFD-based method for modeling bioreactors is the EEM, particularly with regard to bioreactors that have larger sizes and higher volume fractions of solids. The EEM treats the biotic phase and abiotic phase as co-existing interpenetrating continua. The EEM is much less computationally expensive than the ELM because the state of each particle is not calculated separately [144]. Elqotbi et al. incorporated the Monod equation for *Aspergillus Niger* gluconic acid production into CFD using the EEM, and they demonstrated that an increase in viscosity resulted in a decrease in oxygen mass transfer, which further affected gluconic acid production [145]. Morchain et al. used the EEM to simulate a fermentation process in a 70 L bioreactor and an industrial-scale 70 m^3^ bioreactor to explain the performance degradation of industrial bioreactors that mix poorly [146]. Du et al. used the EEM to establish a CFD model coupled with an unstructured kinetic model to explore the effect of bioreactor size and operating conditions on the fermentation process, and they were able to scale-up DHA fermentation from a 5 L bioreactor to a 35 m^3^ bioreactor [49]. Jing et al. also built a coupling model to fine-tune the biological production process to a reasonable extent, and they were able to scale-up the production of ferulic acid into vanillin, from shaker to bioreactor, with a conversion rate up to 94% [147].

**Table 3 bioengineering-09-00473-t003:** Overview of the applications of different kinetics and computational fluid dynamics coupling frameworks used during fermentation.

Approach	Application	Refs.
ELM	The transcriptional changes of *Clostridium ljungdahlii* cells subjected to CO restriction in a 125 m^3^ bubble column bioreactor was predicted, which guided the scaling-up of production.	[140]
ELM	The decrease in penicillin production when using *P. chrysogenum* due to glucose gradient in a 54 m^3^ stirred tank reactor was predicted.	[57]
ELM	The formation of population heterogeneity in *E. coli* in a 54 m^3^ bioreactor was predicted.	[141]
ELM	The difference in microalgae biomass in different photoreactors caused by different light distributions was predicted.	[143]
EEM	The reason for the decrease in the gluconic acid yield during the production of gluconic acid by *Aspergillus Niger* was revealed, which was due to the decrease in oxygen mass transfer due to the increase in medium viscosity during fermentation.	[145]
EEM	The performance degradation of the industrial bioreactor under poor mixing conditions was explained by comparing the flow field environment of the laboratory bioreactor (70 L) with that of the industrial (70 m^3^) bioreactor.	[146]
EEM	The effects of the size of the bioreactor and the operating conditions on DHA fermentation were predicted, and DHA fermentation was scaled up from 5 L to 35 m^3^.	[49]
EEM	The biological production process was fine-tuned by coupling CFD and biokinetics, and the scale required to turn ferulic acid into vanillin (scaling it up from shaker to bioreactor) was realized, with a conversion rate up to 94%.	[147]

As previously indicated, computational frameworks that combine biological models and CFD models can provide information on metabolic reactions induced by environmental fluctuations in industrial bioreactors; thus, they can indicate what improvements to make with regard to the design of bioreactors and the fermentation process. One of the main obstacles in the adoption of coupled frameworks is the computing time required, but technological advancements, especially in GPU-based computing, may make real-time simulations feasible [137,148]. In the future, coupled frameworks will be combined with visualizations, such as virtual reality, to allow operators to ‘see’ what’s going on inside bioreactors [149,150,151].

## 6. Conclusions and Future Perspectives

Biomanufacturing is a research hotspot in the modern manufacturing industry, which can transform renewable raw materials into value-added compounds through microbial fermentation. In addition to developing engineered strains that can act as mini-factories, optimizing the fermentation process and scaling-up production are also essential factors in improving the productivity of biomanufacturing. Fortunately, today, several modeling methods are available to build mathematical models of the fermentation process in order to guide the development of strategies for improving fermentation efficiency. This paper briefly summarizes two important modeling approaches for building fermentation models: mechanistic modeling and data-driven modeling.

The earliest and most commonly used mechanistic models are kinetic models, which can be used to describe the macroscopic dynamic characteristics of fermentation process. At present, they still play an important role in biomanufacturing processes such as bioethanol and surfactants. With the advancement of omics technology, biologists now have a thorough understanding of how cells function, allowing them to build more microscopically detailed mechanistic models, or so-called CBM models. In comparison to kinetic models, CBM models provide a more comprehensive representation of complex cellular factories, providing quantitative and more detailed predictions to aid in the development of fermentation strategies. CBM can be subdivided into several specific methods, including MFA, FBA, and DFBA, which can quantify metabolic flux distributions in cellular metabolic networks. Unlike mechanistic models, data-driven models use mathematical knowledge to analyze fermentation data in order to guide process optimization. The development of modern computer technology, especially ML technology, has enriched the application scenarios of data-driven models, which can now describe the fermentation process more accurately, and can ignore the internal metabolism of cells. Data-driven models are widely used in predicting fermentation results, predicting optimal fermentation conditions and controlling processes. In addition, the development of omics technology and ML technology also provides a boost for the development of mechanistic models. Many researchers tend to use ML to interpret large-scale datasets and analyze metabolic networks, in addition to constructing a hybrid model of ML and CBM synergism. Finally, an integrated tool involving the coupling of biological models with CFD models, for model-based culture behavior prediction and model-aided bioreactor operation design, to be used during the process of scaling-up bioreactors, was discussed.

In our view, after collaborating with a range of disciplines to develop new mathematical and computational tools, while enhancing our knowledge of biology, the evolution of fermentation process modeling could be limitless. In the future, it will be possible to dynamically reflect fermentation changes using simulation models, and it will be possible to regulate the fermentation process by digitally controlling it online.

## Figures and Tables

**Figure 1 bioengineering-09-00473-f001:**
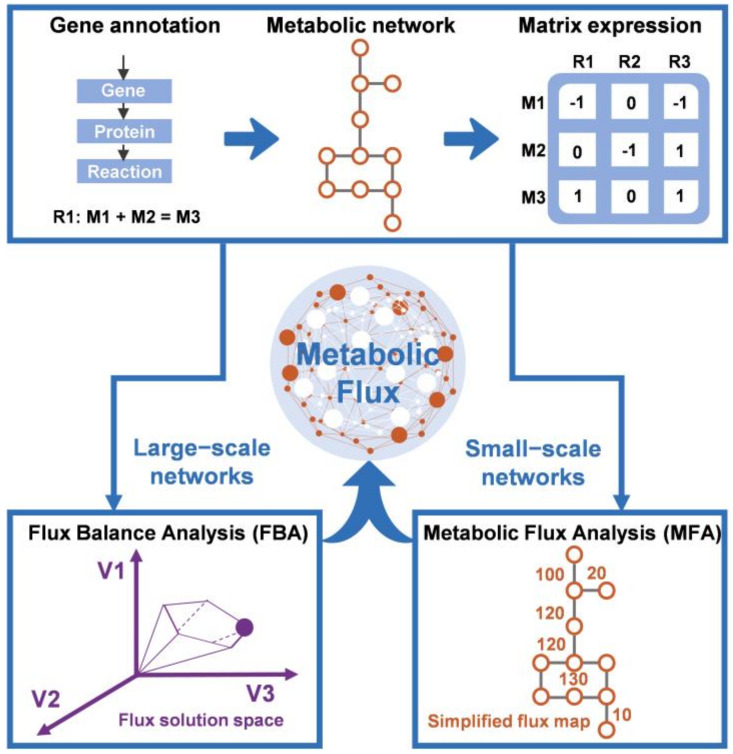
Schematic of constraint-based modeling (CBM) methods.

**Figure 2 bioengineering-09-00473-f002:**
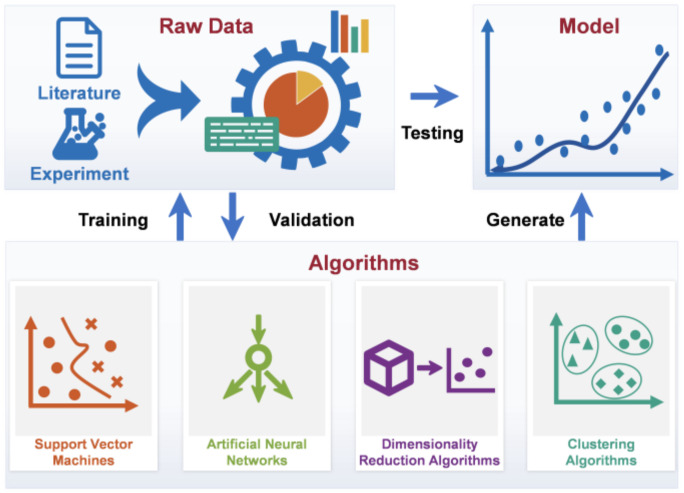
Data-driven modeling steps.

**Figure 3 bioengineering-09-00473-f003:**
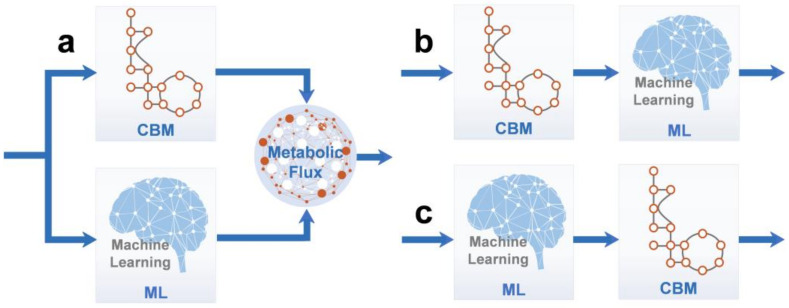
Different forms of CBM and ML integration. (**a**) Processing omics data and predicting parameters using ML. (**b**) Obtaining more biological insights from the metabolic flux date using ML. (**c**) Processing omics data using ML, which is then used as input data to construct CBM.

**Figure 4 bioengineering-09-00473-f004:**
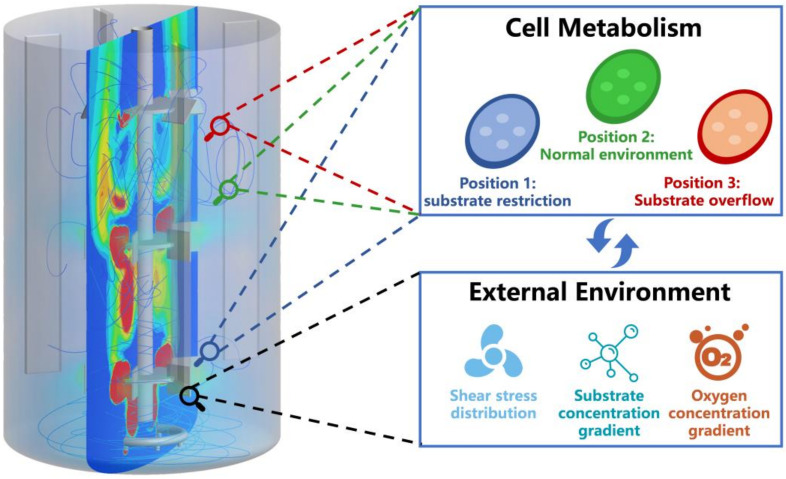
Schematic of metabolic models and computational fluid dynamics coupling.

**Table 1 bioengineering-09-00473-t001:** General macroscopic kinetic equations.

Name	Expression	Function	Refs.
Monod Kinetics	$\mu =\mu_{max}\frac{\left[S\right]}{K_{S}+\left[S\right]}$	To describe microbial growth based on the consumption of one substrate.	[48]
Double Monod Kinetics	$\mu =\mu_{max}\frac{\left[S_{1}][S_{2}\right]}{\left(K_{s1}+[S_{1}])(K_{s2}+[S_{2}]\right)}$	To describe microbial growth based on the consumption of multiple substrates.	[49]
Enzyme inhibition Kinetics	$\mu =\mu_{max}\frac{\left[S\right]}{K_{s}\left(1+\frac{I}{K_{I}}\right)+\left[S\right]}$	To describe microbial growth in the presence of competitive substrate inhibition.	[50]
Contois Kinetics	$\mu =\mu_{max}\frac{\left[S\right]}{K_{S}[S_{0}]+\left[S\right]}$	To describe microbial growth in a high-density culture.	[51]
Powell Kinetics	$\mu =(\mu_{max}+m)\frac{\left[S\right]}{K_{S}+\left[S\right]}-m$	To describe microbial growth while considering the basal metabolic consumption of cells (e.g., metabolite turnover).	[52]
Moser Kinetics	$\mu =\mu_{max}\frac{{\left[S\right]}^{n}}{K_{S}+{\left[S\right]}^{n}}$	To describe microbial growth in situations where cells have multiple pathways to utilize substrates.	[53]
Logistic Equation	$\mu =\mu_{max}(1-\frac{\left[X\right]}{K_{X}})$	To describe microbial growth without any biological explanation other than the assumption that there is a maximum cell growth concentration.	[41,42]
Haldane–Andrew Model	$\mu =\mu_{max}\frac{\left[S\right]}{K_{s}+\left[S\right]+\frac{{\left[S\right]}^{2}}{K_{i}}}$	To describe microbial growth while considering that some substrates are toxic to cells and can inhibit cell growth at high concentrations.	[39,52]
Diauxic Growth	$\mu =\mu_{m1}\frac{\left[S_{1}\right]}{K_{S1}+\left[S_{1}\right]}+\mu_{m2}\frac{\left[S_{2}\right]}{K_{S2}+[S_{2}]+\frac{{\left[S_{2}\right]}^{2}}{K_{i}}}$	To describe microbial growth while considering that there are two carbon sources, S1 and S2, during cell growth and that the cell preferentially uses S1.	[54,55]
Luedeking–Piret Equation	$\frac{dP}{dt}=\alpha \frac{d\left[X\right]}{dt}+\beta \left[X\right]$	To describe the production rate of product P in the case where product synthesis is related to the growth rate and cell density of microbial cells.	[54,55]

μ: the specific growth rate of a microorganism; $\mu_{max}$: the maximum specific growth rate of a microorganism; $K_{S}$: the Monod constant; [*S*]: substrate concentration; [$S_{0}$]: initial substrate concentration; [*X*]: biomass concentration; I: competitive inhibitor; $K_{I}$: the Monod constant of competitive inhibitor; m: a term of specific maintenance rate; $K_{I}$: the inhibition constant equal to the highest substrate concentration [*S*] when μ = 0.5$\mu_{max}$; n: number of binding sites of enzyme to substrate *S*.

## Data Availability

Not applicable.

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
