# Peer review of "Optimization and Scale-Up of Fermentation Processes Driven by Models"

_bioengineering, 2022, doi:10.3390/bioengineering9090473_

Round 1

Reviewer 1 Report

The paper presents a summary of the main existing strategies for tackling the complex problem of bioreactor modelling and optimisation. Although, in general, the information presented is well known to researchers working in this field, I think the paper may be useful for those who want a quick introduction into the context of the different options for dealing with these problems.

Therefore, I think the manuscript could be acceptable for publication. I will make only a few minor comments and/or suggestions below.

  1. Between lines 57-60, reference is made to bioreactors being the foundation of bioprocessing, although they are a fundamental part of what a bioprocess is, one should not be so categorical in this statement and forget the importance of upstream and downstream operations as well. In addition, perhaps reference should have been made throughout the paper to the problems of modelling bioreactors in conjunction with that of other fundamental operations in any bioprocess.
  2. I would delete the term true, in line 65.
  3. I would replace the full stop at the end of line 79 with a semicolon.
  4. In lines 81 and 82 reference is made to the fact that this paper will focus on mechanistic models, however an introduction is made to non-mechanistic models as well. Therefore, I suggest modifying this sentence slightly, even though the paragraph starting on line 89 summarises the outline of the manuscript.
  5. In line 114 the term mycelium has been chosen; due to the specific use of this term to refer to fungi, I suggest that it be changed to a more generic term.
  6. In lines 443 and 444, as written, it seems to suggest that this particular work shows that an increase in viscosity leads to a reduction in oxygen transfer, obviously not something new, but in any case, it could be said that they confirmed a well-known fact.

Reviewer 2 Report

The manuscript presents a collection of models that have been used to optimize biotechnological processes. The paper fits into the scope of the Journal but presents incoherent and incorrect interpretation of mathematical models. I cannot see for whom this article is helpful. The text is well written but poorly structured, as the basic structures of the models have not been considered.

The publication should be rejected. The following comments might help to revise.

Abstract

Non-mechanistic modeling strategies, including kinetic models: What is that supposed to be, since kinetic models are mechanistic models.

Introduction

They are mechanistic models and not mechanical models.

End of 2.1

Models are poor for great differences in operation conditions: This actually applies to all models, as these are usually only made and valid for the description of a certain aspect. The question is rather whether these models might be suitable for scale-up purposes.

2.2 data driven and kinetic are fundamentally contradictory

Here, however, the optimal working condition is found with the help of models (mostly via DoE), which has little to do with a kinetic model.

Apparently, the author is not able to present the models and thus their significance for biotechnological processes in a structured way. The question of what the input and what the output of the models are is also never answered. Which data are needed for the model development, which data are needed for the application and what does the model thus do for process optimisation. Without this, the significance of the models for the processes cannot be satisfactorily classified.

Reviewer 3 Report

The paper presents a review of models and modelling techniques applied to biological production of selected chemicals and their benefit for process optimization and scale-up.

I completely agree with the core idea of the paper, that models and model-based tools are valuable ingredients in bioengineering that can save large amount of costs, time and improve performance. The paper is well written and touches the main aspects claimed in the title.

In my view, however, the paper adds little new insights. It is a collection of well-known facts and a list of widely used techniques presented at a very basic level. Except for the main message, with which I agree, I am not sure how the paper will be actually useful to potential readers.

Suggestions:

The authors might wish to enhance the abstract by specifying to which readers the paper would be useful and in which way

The term “mechanical” usually refers to “mechanics” as a branch of physics. I think most of the time the authors mean “mechanistic” in the sense of “describing the inner phenomena”. Please replace throughout.

In my view CFD is a modelling technique and CFD models are a category of models. I guess by “coupling models with CFD” the authors mean “coupling biological models with fluid dynamics models”. Please rephrase where needed.

line 47: WILL lead to failure: perhaps MAY lead to failure

Round 2

Reviewer 2 Report

The manuscript presents a well structured collection of models that have been used to optimize biotechnological processes. The paper fits into the scope of the Journal and presents new approaches and combinations of different modelling techniques with concrete examples of application. The text is well written and clearly structured.